# Political Trust and Festival Attachment: Influencing Residents’ Engagement in Traditional Festivals

**DOI:** 10.3390/bs13090741

**Published:** 2023-09-05

**Authors:** Jing Zhang, Guangquan Dai

**Affiliations:** 1Department of Economics and Management of Convention and Exhibition, South China University of Technology, Guangzhou 510000, China; jingjing84322@sohu.com; 2Tourism and Air Service College, Guizhou Minzu University, Guiyang 550001, China

**Keywords:** festival attachment, political trust, traditional festivals, participation willingness, Hmong (Miao) New Year

## Abstract

Traditional festivals hold immense cultural and tourism value, but striking a balance between preservation and adaptation in the face of globalization is challenging. This study focuses on the Hmong New Year, the largest traditional festival in a prominent Hmong settlement in China. Through mixed research methods, it examines the impact of festival attachment and political trust on residents’ attitudes toward festival tourism development. The results reveal the significant influence of festival attachment on residents’ perception and judgment of festival tourism, with political trust playing a crucial moderating role. Successful repetition of festival activities fosters stable cognitive perceptions of festival tourism, outweighing potential risks. This research enhances our theoretical understanding of festivals and provides insights into the sustainable development of traditional Hmong festivals, offering support for studying traditional festivals in diverse cultural contexts.

## 1. Introduction

Traditional festivals are an integral part of the cultural heritage of diverse societies, deeply ingrained in collective memory and cultural traditions. In today’s context, these festivals have gained prominence as valuable cultural tourism assets, wielding substantial influence in various nations. The preservation of traditional festival culture has emerged as a shared cultural imperative within the tourism domain. Community residents play a pivotal role in realizing cultural preservation and sustainable tourism development [1,2]. Community support not only enhances the shared experience for residents and visitors [3], but also mitigates the risks associated with potential resistance, such as obstruction, opposition, and legal advocacy [4,5]. However, this endeavor faces multifaceted challenges in the wake of globalization and industrialization. Striking a balance between preserving the cultural uniqueness and diversity of traditional festivals while navigating the forces of globalization and industrialization represents a critical issue in modern society.

This research seeks to address the complex challenge of developing traditional festivals within the context of globalization and industrialization while safeguarding their cultural richness and diversity. Since the early 20th century, governments at all levels in China have actively engaged in the reconstruction of local traditional festivals to align them with national aspirations. They have employed policies, laws, campaigns, and rituals to protect cultural diversity and shape collective identity. These emerging traditional festivals are a product of the state’s involvement in cultural construction. In China’s autonomous minority regions, local governments have integrated traditional festivals into the statutory festival system to preserve cultural traditions, foster national identity, and promote multi-ethnic unity and integration. However, the emotional recognition and active participation of cultural natives in these reconstructed traditional festivals have become essential factors influencing the preservation of traditional minority festival culture and the sustainable development of local cultural tourism.

While existing studies on resident attitudes in tourism communities have predominantly relied on the social exchange theory (SET) [6], which emphasizes perceived value and costs as critical motivators [7], the role of personal emotions in influencing human judgment, decision-making, and behavior is well-recognized in psychological research [8]. These emotions develop cumulatively through the assessment of events over time. The economically rational and systematic approach of SET tends to overlook the emotional appeal experienced by community residents, limiting its explanatory power in understanding residents’ willingness to participate in and support tourism activities. In contrast, this research introduces a novel model based on affective events theory (AET) to examine how recurring festivals influence residents’ motivation and behavior through emotional and economic value judgments. This theory systematically elucidates the mechanisms by which emotional responses affect attitudes and behaviors through the complete chain of “events-emotions-attitudes-behaviors”, as illustrated in Figure 1.

Russell Dalton’s definition of “trust” is an affective orientation, wherein individuals believe that political entities act with deliberate goodwill and are capable of actively serving the public’s interests and needs [9]. Political trust emerges as a crucial gauge for predicting resident attitudes and their support for the tourism industry [4,10]. Interestingly, certain studies have uncovered instances where residents continue to endorse the hosting of large-scale events, even in the presence of mistrust towards political individuals and institutions [11]. This paradoxical observation implies the presence of alternative factors, some of which may mitigate the correlation between trust, perceived impact, and support.

Despite its profound significance, festival research frequently neglects the dimension of political trust. Consequently, this research introduces a comprehensive framework that amalgamates the affective events theory (AET) with the inclusion of political trust, accentuating the pivotal roles of emotions and governmental involvement in comprehending and promoting resident participation and support in tourism activities. By examining the intricate interplay between emotions, economic value judgments, and resident behavior, this study significantly advances the existing body of literature and contributes valuable insights into the multifaceted dynamics underlying resident attitudes toward tourism.

This study aims to explore how political trust, festival attachment, perceived value, and perceived cost collectively shape residents’ willingness to participate in traditional festivals from the perspective of community residents, unraveling the complex transmission mechanisms involved. The study adopts AET as a research framework (Figure 1) to elucidate the factors and mechanisms influencing residents’ participation willingness in traditional festivals during their evolution. The findings will provide new ideas and theoretical support for preserving traditional festival culture and achieving the sustainable development of tourism destinations.

This study focuses on the Hmong New Year festival, celebrated in a Hmong village community located in Leishan County, Guizhou Province, China. It aims to provide a comprehensive case study of tourism development in traditional festivals of ethnic minorities under Chinese governance. The study makes significant theoretical contributions in two main areas: firstly, this study highlights the intrinsic cultural and emotional significance of traditional festivals, surpassing their economic and utilitarian aspects. By emphasizing the cultural and emotional dimensions, it enriches the theoretical framework of emotional factors in the field of traditional festival tourism research. Consequently, it offers a fresh and comprehensive perspective for understanding the tourism value of traditional festivals, considering their cultural heritage and emotional experiences. Secondly, the study develops a comprehensive model that illuminates the intricate relationship between emotional events and residents’ willingness to participate in traditional festival tourism. Through the application of a moderated mediation model and the inclusion of participatory observations of festivals, the research unveils the crucial moderating role of political trust. It demonstrates how political trust influences the connection between perceived value, festival attachment, and residents’ participation intention. This novel finding provides valuable insights and theoretical support for the advancement of traditional festival tourism research, expanding the theoretical scope and deepening our understanding of the social factors that shape tourism decision-making processes.

### 1.1. Literature Review and Hypotheses

#### 1.1.1. The Influence of Residents’ Perceived Value on Their Participation Willingness in Festivals

Numerous studies have shown that festivals and public events play an important role in community development [12]. It includes increasing the international visibility and reputation of the event host, triggering a surge in visitor numbers, increasing business and employment opportunities in the destination [13], and even creating new business opportunities [14].

Residents’ perceived value stems from a comprehensive judgment of the impact of an event. It has been well documented in the tourism literature that residents’ perceptions of the impact of an event are key in shaping their attitudes [13,15,16]. Nunkoo’s study found that tourism development support was positively related to residents’ perceived benefits and negatively related to residents’ perceived costs [17]. In contrast, tourism research in China has found that when residents’ perceived benefits outweigh the costs, they tend to participate in and support tourism activities and tolerate the negative impacts of tourism development [18]. Some reports have even found that residents do not perceive the generation of environmental damage or costs as the main problem of the event [19,20]. Differences in the degree or stage of tourism development may lead to differences between residents’ perceptions and attitudes towards tourism impacts [21,22].

In contrast, studies based on the context of the event will focus more on the impact generated by the event and are more likely to detect the dynamics of changes in residents’ attitudes and perceptions. Several event-based studies have found that residents’ assessments of the impact of the event change over time [11,23]. The actual living environment and perceptions of residents in festival host communities are influenced not only by a single event but also by the cumulative impact of events that recur over the years [24]. Some studies have even found that residents tend to selectively forget the negative effects of hosting events and only remember positive experiences [25].

Overall, traditional festivals are annual recurring events, and participation willingness is an indicator of an individual’s motivation to participate and can predict their participation behavior in advance. Therefore, this study proposes the following hypotheses based on social exchange theory: 

**Hypothesis** **1** **(H1).**
*Perceived festival value has a positive and significant effect on residents’ participation willingness.*


**Hypothesis** **2** **(H2).**
*Perceived festival costs have a negative and significant effect on residents’ participation willingness.*


#### 1.1.2. The Impact of Festival Attachment on Residents’ Participation Willingness in Festivals

Emotions play a crucial role in understanding human behavior [26,27]. The affective events theory (AET), developed by Howard M. Weiss and Russell Cropanzano in 1996, explores the relationship between emotional reactions to affective events experienced by organizational members in their work and their attitudes and behaviors. This theory identifies various factors that lead to emotional reactions in individuals and elucidates how these emotional reactions impact their attitudes and behaviors. Emotional responses triggered by events exhibit a multidimensional structure, and AET examines the structure of emotional responses to events [8]. AET posits that individuals’ emotional responses are activated by their experiences of events, and these emotional responses subsequently influence their attitudes and behaviors [28]. These emotional responses can affect behavior through two pathways: affect-driven behaviors, which are directly guided by emotional responses, and judgment-driven behaviors, which indirectly influence behavior by shaping attitudes [8].

Festival attachment refers to an emotional bonding relationship where participants internalize the event as part of their self-concept and develop a strong attachment to the event [29]. This emotional connection generates memories and associations related to the event [30], resulting in heightened attention and motivation to participate in the festival [31,32]. Research has shown that event attachment behaviors moderate the perception of benefits and costs, thereby influencing residents’ attitudes and behaviors [33]. Residents who possess a high level of emotional attachment to their local community are more likely to perceive the positive impacts of tourism and overlook its negative effects [4]. Increased attachment levels may encourage residents to focus on the positive functional value generated by the event, thereby motivating them to develop supportive behaviors and attitudes.

The findings suggest that festival attachment plays a pivotal role in influencing residents’ willingness to participate in traditional festivals and their perceived value of such festivals, supporting the following hypotheses:

**Hypothesis** **3a** **(H3a).**
*Festival attachment has a positive and significant effect on residents’ willingness to participate in traditional festivals.*


**Hypothesis** **3b** **(H3b).**
*Festival attachment has a positive and significant impact on the perceived value of festival tourism.*


**Hypothesis** **3c** **(H3c).**
*Residents’ perceived value mediates the relationship between festival attachment and their willingness to participate.*


#### 1.1.3. The Moderating Effect of Political Trust on Participation Willingness: The Role of the Perceived Value of Traditional Festivals and Festival Attachment

Political trust refers to people’s confidence and trust in their government or political institutions, and their belief that these entities will adopt policies and measures that meet their interests or give results that better meet their expectations [34]. Political trust is an important indicator that predicts residents’ attitudes toward tourism development in many empirical studies [15,35].

Residents’ trust in government increases their confidence in the social exchange process and stimulates their participation willingness in the exchange [34]. Studies have shown that residents with high trust in government are more likely to support large events organized by the government as they believe these events can bring them greater benefits [16,36]. Conversely, a lack of trust in government can make residents more concerned about the possible negative effects of festivals, leading to a decrease in their participation willingness [37,38]. Furthermore, studies have found that residents’ festival attachment can affect their supportive behavior toward the government, with high festival attachment replacing low political trust and contributing to positive attitudes and behaviors toward hosting large festivals [15,39].

In summary, the above analysis suggests that political trust may influence the relationship between residents’ perceived value of festival events, festival attachment, and participation willingness. Therefore, this study proposes the following hypotheses to test how political trust affects residents’ participation willingness:

**Hypothesis** **4a** **(H4a).**
*Political trust has a significant positive effect on perceived value.*


**Hypothesis** **4b** **(H4b).**
*Political trust has a significant negative effect on perceived costs.*


**Hypothesis** **5a** **(H5a).**
*Political trust moderates the relationship between residents’ perceived value and participation willingness.*


**Hypothesis** **5b** **(H5b).**
*Political trust moderates the relationship between festival attachment and participation willingness.*


As shown in Figure 1, a conceptual framework was developed based on all of the above discussion. The model illustrates the various factors that influence residents’ participation willingness, as well as the possible mechanisms of action between them.

## 2. Research Methodology

### 2.1. Questionnaire Design and Measures

The survey instrument was developed through an extensive literature review and tailored for the event context. A rigorous double-translation procedure was used to translate the initial questionnaire into Chinese, and experts provided feedback to improve language, semantics, and content. A small-scale pre-test with 65 Hmong student volunteers was conducted to improve clarity, followed by an exploratory factor analysis to eliminate invalid items. The final 24-item questionnaire had excellent reliability with a Cronbach’s alpha coefficient of 0.873. Statistical testing, including KMO and Bartlett’s chi-square tests, discrimination testing, and *t*-tests, were conducted to assess the data’s suitability. Participants provided feedback on the revised questionnaire, finding it clearer and more comprehensible.

Perceived value was measured using three items adapted from Prayag (2013) and Gursoy (2006) [4,14]. Event attachment was measured on a four-item scale adapted from Filo (2014) and Ouyang Z (2017) [33,40]. Participation willingness was measured using four items from Rasoolimanesh (2017) [41]. Political trust was measured using four items adapted from Robin Nunkoo and Ouyang Z (2017) [33].

This study included three control variables: length of residence, source of income, and education level, based on previous research [35,42]. The questionnaire comprised two parts: the first part measured latent variables with a total of 24 items on a five-point Likert scale. The second part collected primary demographic data such as gender, age, ethnicity, income, occupation, duration of residence, and place of residence. These established scales have demonstrated good reliability and validity in previous studies conducted by domestic and international scholars.

### 2.2. Research Background

Leishan County is located in southeastern Guizhou Province, China, and is a cultural center for the Hmong community. The county has a total population of 134,000, with Hmong people accounting for 83.2%. The Hmong New Year celebration, which dates back over 5000 years and honors their ancestor Chi You, is an important cultural event and national intangible heritage of China. This festival marks the end of the old year and the beginning of a new one, lasting from three to fifteen days, with different Hmong clans celebrating at varying times. To boost tourism development, the Leishan County government launched the first Chinese Leishan Hmong New Year Festival in 2000, which was followed by the Leishan County Hmong New Year Cultural Festival. This cultural festival aims to attract tourists while preserving Hmong cultural traditions.

Through the successful promotion of the Hmong New Year Cultural Festival, Leishan County has become an important brand for Hmong cultural tourism in China. In 2019, Leishan County received 13,219,500 tourists, generating comprehensive tourism revenue exceeding 11.897 billion yuan. Moreover, Leishan County’s rural tourism-based approach to poverty alleviation was recognized as an outstanding poverty reduction case by the World Tourism Alliance, leading to its removal from being a national poverty-stricken county in 2019.

### 2.3. Data Acquisition and Analysis Methods

This study primarily investigated community residents participating in the Hmong New Year festival activities. The research commenced with small-scale in-depth interviews in June–July 2021 among community residents in Xinqiao Village of Danjiang Town and Langde Town in Leishan County, Qiandongnan, Guizhou. Subsequently, building upon the insights from the interviews, a large-scale questionnaire survey was conducted during the Hmong New Year period from October to December 2021 in Danjiang Town, prominent tourist villages in Langde Town (Zailangde Upper and Lower villages), Xinqiao Village of Datang Town, Zhadao Village, and the Xijiang Scenic Area (comprising 8 villages) in Leishan County. In light of the substantial outmigration of residents for employment opportunities and the consequent population decline in some villages, which has led to a relatively low population density, a larger proportion of elderly individuals and children remain in these areas. Each natural village community typically comprises between several dozen to around a hundred households. Consequently, a convenience sampling method was employed in our questionnaire survey.

In this study, three local university students proficient in the Hmong language were recruited to assist in the research. The questionnaire survey utilized a combination of random interception and household surveys, with one representative from each household invited to respond. A total of 500 questionnaires were distributed, with 486 returned. After removing invalid questionnaires (those with missing key variables, contradictory responses, or selecting the same option for all items), a total of 451 valid questionnaires were retained. This yielded an effective questionnaire response rate of 92.8%.

Of the valid sample, 48.12% were male and 51.88% were female, and the surveyed population was predominantly Hmong, with 88.91% identifying as such. The age distribution was mainly concentrated in the 36–45 age group, accounting for 42.35%, followed by 29.71% in the 46–59 age group. In terms of education, university/college, high school/junior college, and junior high school and below were the top three categories, accounting for 43.09%, 36.59%, and 36.59%, respectively. Non-tourism-related industries accounted for 72.51% of the income source, and the top three lengths of residence in the local area were 8–12 years (12.20%), 12–20 years (31.93%), and more than 20 years (47.45%). The sample was highly reliable and representative, with gender, age, marital status, and ethnicity distributions broadly corresponding to the local situation.

## 3. Results 

### 3.1. Validity and Reliability

The questionnaire used in this study demonstrated good reliability, as indicated by Cronbach’s alpha coefficients ranging from 0.817 to 0.916, as assessed by SPSS 26. Exploratory factor analysis revealed a KMO value of 0.89 and a significant Bartlett’s test of sphericity. The analysis resulted in five factors with eigenvalues greater than one, collectively explaining 68.645% of the variance. These findings support the construct validity of the questionnaire.

To address potential standard method bias effects due to self-reported data, the study employed the ‘Controlling for Effects of an Unmeasured Latent Methods Factor’ method [43]. Adding the standard methods factor (CM) to the five-factor model yielded a poor fit, indicating that standard methods bias had only a minimal impact on the study (Table 1). Additionally, the Harman one-way method was used to test for common method bias, and the first factor accounted for only 25.348% (less than 40%) of the unspun axis, suggesting that standard method variance (CMV) had minimal influence.

### 3.2. Confirmatory Factor Analysis and Structural Equation Modelling

Prior to conducting the structural equation analysis, a confirmatory factor analysis (CFA) was performed using Amos 26 to assess the overall model fitness. Standardized loadings for each dimension were examined and presented in Table 2, ranging from 0.711 to 0.91, all reaching significant levels (*p* < 0.001). The CRs for the dimensions ranged from 0.818 to 0.917, exceeding the recommended threshold of 0.7, and the AVEs ranged from 0.53 to 0.689, also exceeding the recommended threshold of 0.5, indicating good convergent validity of the measured dimensions. To further test discriminant validity, the square root of each latent variable’s AVE value was compared to the correlation coefficient between it and its variables, demonstrating high discriminant validity.

In addition, the model fit indices were evaluated: χ^2^ = 279.121; dƒ = 242; χ^2^/dƒ = 1.153; RMSEA = 0.018; GFI = 0.952; AGFI = 0.940; RFI = 0.946. All indicators demonstrated that the model had a good overall fit and was deemed appropriate for subsequent data analysis.

### 3.3. Correlation Analysis

The results of the descriptive statistics and correlation analysis of the variables indicated that the correlation coefficients between the two variables ranged from 0.186 to 0.457, all of which were significant, except for the correlation coefficients between political trust and perceived value and perceived cost, which ranged from 0.024 to 0.106 (Table 3). The correlation coefficient between perceived cost and participation willingness was 0.04, which did not reach a significant level. Therefore, there is no significant relationship between political trust and perceived value and perceived cost, and Hypotheses H4a and H4b are not supported. Additionally, there is no significant relationship between perceived cost and participation willingness, and Hypothesis H2 is not supported. Other partial hypotheses in this study were tentatively confirmed.

### 3.4. Hypothesis Testing

#### 3.4.1. Assessment of The Structural Model

A structural equation model was used to test the hypotheses in this study using maximum likelihood estimation. The results showed that χ^2^ = 333.643, dƒ = 245, χ^2^/dƒ = 1.362, RMSEA = 0.028, GFI = 0.944, NFI = 0.944, CFI = 0.984, and NNFI = 0.982. All fit indices of the model met the predetermined criteria, indicating a good overall fit of the model and its appropriateness for subsequent analysis. The results of the study were also confirmed through hypothesis testing using structural equation path analysis (Table 4).

#### 3.4.2. Analysis of Moderating and Mediating Effects

To address the potential neutralization of mediating effects [44] when moderating variables exhibit different effects at high and low levels, the Process procedure developed by Hayes (2017) was utilized to construct a mediating model incorporating moderation, aligning with the research hypotheses. The findings of this study indicate that the relationship between festival attachment and participation willingness, as well as the relationship between perceived value and festival attachment, are moderated by political trust. Data analysis involved the use of Model 15 of the moderated mediation model within Process [45], with the mediation effect assessed using the Bootstrap method. To mitigate issues of multicollinearity, mean-centering was applied to the variables associated with the moderating effect due to the smaller sample size of 451 participants (while a recommended minimum of 500 was suggested, a sample size of 5000 was considered).

The findings indicated that political trust exhibited a significant negative moderating effect on the association between festival attachment and participation willingness (B = −0.258, t = −5.495, *p* < 0.001). Furthermore, the control variable of educational attainment demonstrated a negative correlation with participation willingness (β = −0.177, t = −2.647, *p* = 0.008). Perceived value was identified as a mediating factor, with effect sizes greater than 0 observed at both low and high levels. The Bootstrap standard errors, confidence intervals, and sampling intervals consistently excluded 0, indicating a consistent mediating role without a moderating effect (Table 5).

#### 3.4.3. Analysis of Mediating Effects

The causal stepwise regression method developed by [46] was employed to examine the mediating effect, and a parallel mediating effect test was conducted with 5000 Bootstrap iterations to support the findings. The mediating effect value of ab was calculated as 0.074 (95% Boot CI: 0.039–0.103), indicating statistical significance. Additionally, the direct effect c’ was found to be significant, and the partial mediating effect accounted for 14.908% of the total effect. Therefore, Hypothesis H3c received support from the data. The detailed results can be found in Table 6, which is presented below.

#### 3.4.4. Analysis of Moderating Effects

##### The Moderating Effect of Political Trust in Festival Attachment and Participation Willingness

A moderation analysis was conducted to investigate the moderating effect of political trust on the relationship between Festival attachment and Participation willingness. Three models were developed: Model 1 examined the direct effect of Festival attachment on Participation willingness without considering political trust, while Model 2 included political trust as a moderator variable. Model 3 further incorporated the interaction term between Festival attachment and political trust.

The results indicated that Festival attachment had a significant impact on Participation willingness (t = 10.901, *p* < 0.001), suggesting its influential role in determining participation intentions. Comparing Model 2 with Model 3, the significant change in the F-value and the presence of a statistically significant interaction term (t = −5.819, *p* < 0.001) indicated that the magnitude of the moderating variable (political trust) varied significantly at different levels of Festival attachment, influencing the willingness to participate.

To further explore the nature of this interaction, a simple slope analysis was conducted. The results revealed that for individuals with low political trust (−1 standard deviation), the effect of Festival attachment on Participation willingness was stronger (B = 0.686, t = 11.436, *p* < 0.001, 95% CI = 0.569–0.804). Conversely, for those with high political trust (+1 standard deviation), the effect was weaker (B = 0.215, t = 3.767, *p* < 0.001, 95% CI = 0.103–0.327). These findings provide support for Hypothesis H5b, which posited political trust as a moderator in the relationship between Festival attachment and Participation willingness. For a visual representation of the results, please refer to Simple Slope Figure 2a.

##### The Moderating Role of Political Trust in Perceived Value and Participation Willingness

The results indicate that the independent variable (Perceived Value) demonstrated significance (t = 8.291, *p* = 0.000 < 0.05), indicating that it had a significant influence on the relationship with participation willingness. The change in F-value from Model 2 to Model 3 was statistically significant, and the interaction term between perceived value and political trust in Model 3 was also significant (t = −5.283, *p* = 0.000 < 0.05). This suggests that the magnitude of the moderating variable (Political trust) differs significantly at different levels when perceived value affects participation willingness. The results of the simple slope analysis show that the effect of perceived value on participation willingness is stronger for individuals with low political trust (−1SD) (B = 0.689, t = 10.040, *p* < 0.001, 95% CI = (0.555, 0.824)) and weaker for those with high political trust (+1SD) (B = 0.207, t = 3.161, *p* = 0.002, 95% CI = (0.079, 0.335)). Therefore, Hypothesis H5a, which posits that political trust moderates the relationship between perceived value and participation willingness, was supported. Refer to Simple Slope Figure 2b for additional details.

## 4. Discussion

Firstly, the study findings reveal a significant positive correlation between residents’ perceived value of festival tourism and their willingness to participate, while perceived costs show no significant correlation with participation willingness. This conclusion aligns with the results of previous research [4,47]. However, some studies have found that residents’ willingness to participate in events and their level of support may change throughout event development [22]. At different stages of tourism development, the impact of perceived costs on tourism support and participation willingness among residents may vary [25]. During our interviews, we discovered that for many elderly residents and women in several villages, agriculture serves as their primary source of income. Income from tourism performances plays a crucial role in their livelihoods. A resident in Lande Village stated, “Festival events enhance the reputation of our village and attract more tourists. Women can earn income by selling handicrafts and participating in performances without having to work elsewhere. Each time we perform, we can earn 100 to 200 yuan”. Interviews revealed that the majority of residents do not mind the disruption caused by the influx of tourists during festival periods. This study suggests that in communities at the early stages of tourism development or with limited economic resources, residents may prioritize the positive impacts of festival tourism, such as enhancing local influence, economic benefits, employment opportunities, and improved living conditions. Therefore, they may selectively overlook or accept certain negative effects. This conclusion is consistent with previous research findings [21,48], indicating that even if residents hold negative attitudes towards tourism, it may not necessarily hinder their willingness to participate. Residents still desire to engage in economic activities related to tourism and benefit from them. This demonstrates that economic benefits remain a significant factor driving residents’ willingness to participate in festival events. Furthermore, similar to other research findings, the survey found that most residents deny receiving direct economic benefits from tourism development. However, they still support tourism development due to the perceived indirect social value of tourism, which may result from the multiplier effects and marginal contributions of the tourism industry to other sectors of the economy [49].

Secondly, the study finds that festival attachment is one of the effective predictor variables influencing residents’ willingness to participate in tourism and their perceived value. Some studies suggest that over time, the repeated hosting of events leads residents to form stable attitudes and opinions based on direct experiences [23,50]. Such direct experiences contribute to residents’ tourism knowledge, affecting their understanding of tourism development issues [38]. Traditional festivals are a type of repetitive event with significant social functions. From the year 2000 until the COVID-19 pandemic in 2020, the local government in Leishan County has successfully organized 20 editions of the Hmong New Year Cultural Festival to promote tourism development. This large-scale tourism festival has facilitated local tourism development, contributing to improvements in the local social, economic, cultural, and ecological environment. It has also stimulated the Hmong residents’ identification with their ethnic identity and emotional connection to participating in the Hmong New Year. This unique emotional attachment leads residents to selectively perceive the positive impacts of the festival. Local elites contribute more meaning to the Hmong New Year through social actions and generate more favorable interpretations and evaluations. Even residents from relatively impoverished villages spontaneously raise funds to organize the Hmong New Year event, hoping to attract the attention of the government and tourists through festival promotion, gain more development opportunities for their villages, and even ignore the costs of organizing the festival and the risks brought by the COVID-19 pandemic. Additionally, the study found that perceived value partially mediates the relationship between festival attachment and participation willingness, with the mediation effect accounting for only 14.908%. Perceived costs have no significant relationship with festival attachment and participation willingness. This indicates that a strong festival attachment leads residents to have a more positive willingness to participate in traditional festivals. They are less influenced by judgments based on the rational values of social exchange and tend to focus more on the positive impacts brought by traditional festival activities while neglecting the negative effects of festivals. These findings shed light on the complex interplay between residents’ emotions, economic value judgments, and their willingness to participate in traditional festival tourism.

Thirdly, this case study reveals that political trust does not directly influence residents’ perceived value of the festival but instead moderates residents’ willingness to participate in the Hmong New Year activities through the perceived value of the event. This finding is consistent with prior research by Gursoy, Nunkoo, Ouyang, and others [15,33,36], which has demonstrated that residents’ trust in the government can impact their attitudes toward hosting events. Social exchange is based on mutual trust among stakeholders in exchange relationships [51]. When residents lack trust in the government, they tend to doubt the competence and intentions of government agents, resulting in more negative perceptions of the impacts of events and reduced willingness to participate in and support large-scale activities [15,52]. Comparing samples before and after event hosting, it was found that over time, residents tend to increase their trust in the government’s ability to organize events, strengthen their attachment to the events, and reduce their perception of negative impacts [23]. The case falls within an autonomous ethnic minority region in China, where government agents and organizers of the Hmong New Year activities are primarily local indigenous Hmong people. Residents have a higher level of trust in these agents. The local government has vigorously promoted local ethnic cultural tourism during the Hmong New Year, effectively lifting residents out of poverty and significantly improving their living conditions. The model of ethnic cultural tourism development in this region has become an excellent case study recognized by the World Tourism Alliance. Therefore, residents’ higher trust in government institutions and event organizers encourages residents to have a more positive perception and a higher willingness to participate in the event.

Fourthly, the study finds that residents with high levels of festival attachment and political trust have a higher willingness to participate. However, when the festival attachment is low, residents’ willingness to participate in the festival is more sensitive to the influence of political trust, and the difference in the moderating effect of political trust on participation willingness is not significant when the festival attachment is high. When festival attachment is low, the level of political trust significantly moderates participation willingness. Related research suggests that this phenomenon may be due to residents exhibiting “confirmation bias” [33] or “cognitive dissonance” [53] in the context of large events. Year after year, repeated events create a special emotional connection and cognition among residents. This may lead residents to selectively ignore factors that contradict their cognition. According to Heider’s balance theory, residents with a high emotional attachment to the festival may experience cognitive dissonance and anxiety if they engage in unsupportive or non-participatory behavior due to low perceived value or political trust. They need to readjust their behavior and cognition to achieve psychological balance. Festival attachment may mitigate the impact of political distrust and low perceived value on participation willingness, increase residents’ enthusiasm for participation, and foster a positive attitude and behavior toward relevant policies and institutions.

## 5. Conclusions

### 5.1. Theoretical Implications

This study examines the concentration of Hmong ethnic minorities in China and their grandest traditional festival, the Hmong New Year, in the central dialectal region of China. The research advances the theoretical knowledge of traditional festivals of ethnic minorities in China by presenting special cases and contributing to the festival studies literature.

The study makes a valuable contribution to the field of festival tourism by applying the affective event theory to overcome the limitations of assuming rational human behavior. By investigating the residents’ perceptions of festival tourism’s impact, festival attachment, and political trust, the research sheds light on their willingness to participate in festivals. This study provides a theoretical framework that enhances our understanding of the motivating factors behind community residents’ engagement in traditional festivals and offers insights for achieving sustainable development of festival culture through effective festival governance. By expanding upon existing research in festival tourism, this study lays a solid foundation for comprehending and promoting community involvement in traditional festivals.

Moreover, the study highlights the crucial role of local governments in organizing and managing festivals during the early stages of festival tourism development. It explains how the political trust generated by government action can significantly impact the social exchange processes of residents. By focusing on the impact of political trust on residents rather than cultural preservation, this paper provides theoretical explanations and empirical support for a deeper understanding of the interplay between residents’ government perceptions of tourism impacts, festival attachments, and participation willingness.

To enhance community residents’ participation willingness in traditional festivals sustainably and effectively, the study suggests fostering positive interactions between the government, festival preparation organizations, and residents, nurturing residents’ emotional attachment to traditional festivals, and providing a theoretical basis and case studies for the government and organizations to carry out cultural management of traditional festivals.

### 5.2. Managerial Implication

This study provides actionable insights for festival management that can be put into practice. Firstly, in the early stages of festival tourism development, organizers should prioritize events that promote social welfare and sustainable development values. This includes initiatives to promote economic development, improve the social living environment, enhance community welfare, and preserve cultural heritage. Such measures can improve residents’ perceptions of the positive impact of festivals and enhance the overall visitor experience.

Secondly, festival management should focus on cultivating trust and engagement among residents. Tourism development in many ethnic minority regions of China has led to uneven resource distribution and low levels of community participation. To address these issues, festival management should guarantee community residents’ rights to make decisions and participate in festival activities, ensuring fairness in participation and resource distribution. Differentiated incentives and compensation should be given to different groups of participants, with particular care and support for disadvantaged and low-income groups. By empowering local communities, festival management can foster a sense of ownership and commitment to the festival, leading to greater sustainability and long-term success.

Thirdly, government and organizational decision-makers must prioritize fairness, legality, and cultural sensitivity when carrying out various festival-related activities. Respect for local populations and cultures should be at the forefront of decision-making, with an emphasis on balancing tourism development goals with cultural preservation and community well-being. Residents should be treated as active participants and contributors to festival activities rather than passive spectators. By involving local communities in the planning and execution of festival-related activities, festival management can foster ownership and a sense of pride in the community’s cultural heritage.

By implementing such strategies, festival organizers and management can create a more inclusive, community-centric approach to tourism development that benefits both local populations and visitors alike.

### 5.3. Limitations and Future Research

This study has several limitations that need to be addressed in future research. Firstly, it focuses on a single case study of the Hmong New Year festival in a specific region, which limits the generalizability of the findings. Future studies should explore multi—regional and multi-ethnic cases to increase the interpretation of the findings and promote their generalizability.

Secondly, the local Chinese culture may have influenced the interpretation of the findings, as SET focuses on individual benefit evaluation while collectivism and communal solidarity are more valued in the local Hmong population. As a communal event, traditional festivals emphasize face (Mian Zi), favors (Ren Qing) [54], and the ‘collective honor’ of the village. The local social networks formed by blood and geographic ties in Hmong villages may prolong social exchange relationships or pursue potentially greater reward value. This differs somewhat from SET and social resource theory, which tend to treat social relations as a form of social capital and exchange resource. Cross-cultural research methods could be used in future research to explore the applicability and limitations of SET in diverse cultural contexts.

Thirdly, this study did not consider the trend of changes in residents’ perceptions and attitudes towards festivals over time. Long-term follow-up surveys are necessary to better understand the evolution of residents’ attitudes and guide cultural conservation and management of traditional festivals.

## Figures and Tables

**Figure 1 behavsci-13-00741-f001:**
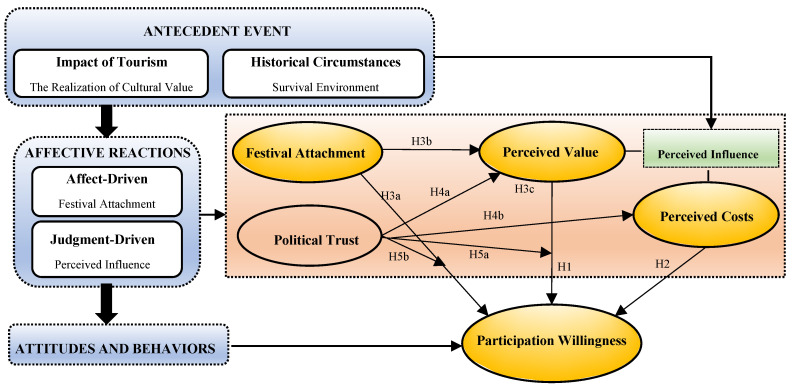
Theoretical Framework Diagram.

**Figure 2 behavsci-13-00741-f002:**
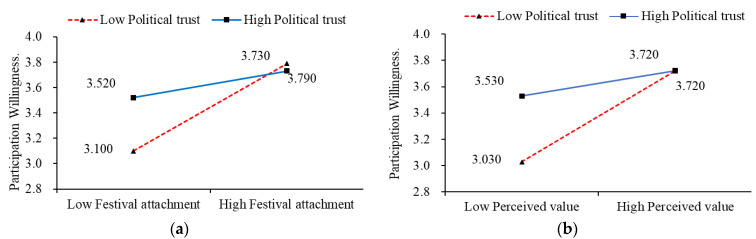
Moderated Effect of Political Trust. (**a**) The Moderating Role of Political Trust in the Relationship between Festival Attachment and Participation Willingness. (**b**) The Moderating Role of Political Trust in the Relationship between Festival Attachment and Participation Willingness.

**Table 1 behavsci-13-00741-t001:** Common Methods Bias Test.

Models	χ^2^	dƒ	GFI	RMSEA	RMR	CFI	NFI	NNFI
One-factor Model	3938.289	252	0.335	0.18	0.202	0.347	0.335	0.285
Five-factor Model	279.742	242	0.953	0.019	0.028	0.993	0.953	0.992
Six-factor Model (Five factors + CM)	7648.155	362	0.513	0.211	0.136	0.524	0.513	0.466

One-factor: Perceived Value + Perceived Cost + Festival Attachment + Political Trust + Participation Willingness. Five-actor: Perceived Value, Perceived Cost, Festival Attachment, Political Trust, Participation Willingness. CM = Common Methods factors.

**Table 2 behavsci-13-00741-t002:** Confirmatory Factor Analysis.

Variable/Item	Standardized Loadings	CR	AVE
Perceived Value		0.891	0.542
PV1. Increased local understanding of history and cultural heritage.	0.705		
PV2. Brought attention to local intangible cultural heritage.	0.73		
PV6. Provided relaxation and entertainment for residents.	0.767		
PV7. Strengthened connections between villages.	0.735		
PV8. Created job opportunities for residents.	0.739		
PV9. Improved public services and infrastructure, enhancing conditions for local economic investment.	0.711		
PV10. Increased local business opportunities.	0.755		
Festival Attachment		0.818	0.53
FA1. Importance of participating in the event.	0.732		
FA2. Sense of identification with the event.	0.706		
FA3. Emotions towards others attending the event.	0.728		
FA4. Perceived importance of the event.	0.742		
Participation Willingness		0.88	0.648
PW1. Willingness to participate in the event again.	0.817		
PW2. Interest in participating in the event’s performance.	0.821		
PW3. Willingness to receive training for participation in the performance.	0.788		
PW4. Willingness to participate in the planning and development of the event.	0.796		
Political Trust		0.87	0.627
PT1. Trust in the correctness of decisions made by relevant authorities in organizing the event.	0.774		
PT2. The belief is that relevant authorities consider the interests of residents when making decisions about the event.	0.787		
PT3. Trust in the reasonableness of the work carried out by relevant authorities in developing the event.	0.783		
PT4. The belief is that relevant authorities have put significant effort into the development of the event.	0.818		
Perceived Cost		0.917	0.689
PC1. Perception of increased pollution due to the hosting of the event.	0.815		
PC2. Perception of increased road closures/disruptions and traffic congestion during the event.	0.91		
PC3. Perception of increased difficulty finding parking during the festival.	0.866		
PC4. Concerns over increased local security risks (e.g., new coronavirus infections, terrorist activities) due to hosting of the event.	0.749		
PC5. Perception of increased visitor harassment and local crime rates (e.g., Hooliganism, disorder, vandalism) due to the event.	0.800		

**Table 3 behavsci-13-00741-t003:** Correlation Analysis.

Construct	Mean	Standard Deviation	1	2	3	4	5
1. Perceived value	3.760	0.657	1.00				
2. Festival attachment	2.997	0.753	0.265 **	1.00			
3. Participation willingness	3.278	0.803	0.364 **	0.457 **	1.00		
4. Political trust	3.151	0.810	0.024	0.186 **	0.326 **	1.00	
5. Perceived Cost	3.449	1.002	0.195 **	0.060	0.042	−0.106 *	1.00

* *p* < 0.05, ** *p* < 0.01, two-tailed.

**Table 4 behavsci-13-00741-t004:** Hypothesis Test Results of Research Structure Model.

Hypothesis	Standardized Coefficients	SE	Z	*p*	Results
**H1** Perceived Value → Participation Willingness	0.270	0.067	5.339	0.000 ***	Supported
**H2** Perceived Cost → Participation Willingness	−0.031	0.034	−0.714	0.475	Not supported
**H3a** Festival Attachment →Participation Willingness	0.472	0.063	8.310	0.000 ***	Supported
**H3b** Festival Attachment → Perceived Value	0.324	0.050	5.421	0.000 ***	Supported
**H4a** Political Trust → Perceived Value	−0.046	0.045	−0.852	0.394	Not supported
**H4b** Political Trust → Perceived Cost	−0.099	0.075	−1.896	0.058 *	Not supported

***: Represents highly significant or very significant, typically indicating that the research results are highly pronounced and have statistical significance. *: Signifies slight significance, typically suggesting that the research results have some degree of statistical significance but not as strong as ***.

**Table 5 behavsci-13-00741-t005:** Hierarchical Multiple Regression.

Variable	Participation Willingness				Perceived Value			
	β	SE	t	*p*	β	SE	t	*p*
Constant	−4.903	0.737	−6.657	0.000 **	2.375	0.262	9.055	0.000 **
Festival attachment	1.203	0.155	7.768	0.000 **	0.222	0.040	5.579	0.000 **
Length of residence	−0.035	0.024	−1.497	0.135	0.031	0.024	1.288	0.198
Education level	−0.177	0.067	−2.647	0.008 **	0.143	0.068	2.096	0.037 *
Source of income	0.016	0.024	0.641	0.522	0.040	0.025	1.608	0.109
Political trust	1.997	0.215	9.289	0.000 **				
Perceived value	1.181	0.169	6.972	0.000 **				
Festival attachment * Political trust	−0.258	0.047	−5.495	0.000 **				
Perceived value * Political trust	−0.261	0.051	−5.070	0.000 **				
*R* ^2^	0.437				0.089			
Adjusted *R*^2^	0.425				0.079			
*F*	*F* (8, 442) = 42.883, *p* = 0.000				*F* (4, 446) = 10.897, *p* = 0.000			

* *p* < 0.05, ** *p* < 0.01.

**Table 6 behavsci-13-00741-t006:** Hierarchical Multiple Regression.

Variable	Model 1	Model 2	Model 3
Constant	2.219 ** (7.484)	2.375 ** (9.055)	1.423 ** (4.615)
Residence Time (RT)	−0.007 (−0.269)	0.031 (1.288)	−0.018 (−0.680)
Education Level (EL)	−0.174 * (−2.269)	0.143 * (2.096)	−0.222 ** (−3.008)
Income Source (IS)	0.024 (0.833)	0.040 (1.608)	0.010 (0.372)
Festival Attachment (FA)	0.498 ** (11.089)	0.222 ** (5.579)	0.424 ** (9.541)
Perceived Value (PV)			0.335 ** (6.545)
*R* ^2^	0.220	0.089	0.288
Adjusted *R*^2^	0.213	0.081	0.280
*F*	*F* (4, 446) = 31.400, *p* = 0.000	*F* (4, 446) = 10.897, *p*= 0.000	*F* (5, 445) = 36.045, *p* = 0.000

* *p* < 0.05, ** *p* < 0.01 (t-values in parentheses). Regression Model 1 represents the regression of the independent variables (Residence time, Education level, Income source, and Festival attachment) on the dependent variable Participation willingness. PW = 2.219 − 0.007RT − 0.174EL + 0.024IS +0.498FA. Regression Model 2 shows the regression of the independent variables (Residence time, Education level, Income source, and Festival attachment) on the mediator variable Perceived value. PV = 2.375 + 0.031RT + 0.143EL + 0.040IS + 0.222FA. Regression Model 3 represents the simultaneous regression of the independent variables (Residence time, Education level, Income source, Festival attachment, and Perceived value) on the dependent variable Participation willingness. PW = 1.423 − 0.018RT − 0.222EL + 0.010IS + 0.424FA + 0.335PV.

## Data Availability

Data are available from the corresponding author on reasonable request.

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
