# Peer review of "Political Trust and Festival Attachment: Influencing Residents’ Engagement in Traditional Festivals"

_behavsci, 2023, doi:10.3390/bs13090741_

Round 1

Reviewer 1 Report

Thank you for the opportunity to review the article “Political Trust and Festival Attachment: Influencing Residents' Engagement in Traditional Festivals”. The article is engaging, exploring the impact of festival attachment and political trust on residents' attitudes toward festival tourism development.

The theoretical background is appropriate for this subject, and the literature review it’s extensive.

Overall, the subject is correctly attributed to the Social Psychology section of the Behavioral Sciences journal, special issue “Psychological Perspectives of Social and Cultural Differences” by studying a contemporary situation and its impact at a local level. 

It is important to emphasise that the article is well-structured, and the results are clearly presented. There is a Discussion section and the limits of the research are presented.

However, I think the authors should present the sampling procedure a bit more in-depth. We know that the volume is 500 questionnaires and that the respondents were randomly chosen, but we do not know from what volume of the population and what the bias is. Also, given that in-depth interviews were conducted before the quantitative study was applied, I think there should be a section in the article describing the study results.

Reviewer 2 Report

This study examines the impact of festival attachment and political trust on residents' attitudes toward festival tourism development. The paper focuses on a very interesting theme. The goal of the paper is clear and is well motivated. The study has the potential to make a useful contribution to the research areas. I have listed my concerns below.

1.This study utilized a mixed-method research design to collect data, however, we cannot find any detail description about the interviews and how the interview questions help to conduct the research. More detail about the method of the paper can be added.

2. There is the absence of a background section/literature review on Affective Events Theory (AET). Such section is necessary to develop an understanding of the key concepts of attitude and behavior ,summarize what we already know on the phenomenon of the research issue, and hence position the paper with respect to existing work.  

3. In the introduction part, the motivation of this paper is not so strong, please revise it with a stronger motivation. For example, what we already know and what we havent known about resident attitudes and behavior, the authors could explain the reason why political trust should be added to the research model.

4. In line 85,there is Leshan County,but in other part of the paper, there is Leishan County.

Reviewer 3 Report

Overall, the work is quite interesting, but some revisions are needed:

1)    Lines 51 “Social Exchange Theory add (SET)

2)    Lines 68-70: It is not at all clear what is the connection between "good political trust and promotion sustainable tourism development and achieving effective festival governance."

3)    Lines 88: not “Firstly” but “firstly”

4)    Lines 136: Overall, T    Overall, t

5)    Since the research objectives and methodology are not clearly explained at the beginning of the paper, it is difficult to appreciate the discussion and the conclusions.

6)    I suggest explaining more clearly the transition from the nine hypotheses formulated to the results; although the methodology is well written, it may be difficult to understand for non-experts.

7)    Since the authors insist a lot on the political trust throughout the discussion, it would be useful if they explained better what has already been indicated in point 2 of my review.
